# Carbon Quantum Dots Based Chemosensor Array for Monitoring Multiple Metal Ions

**DOI:** 10.3390/molecules27123843

**Published:** 2022-06-15

**Authors:** Tianlei Qin, Jiayi Wang, Yuanli Liu, Song Guo

**Affiliations:** Guangxi Key Laboratory of Optical and Electronic Materials and Devices, College of Materials Science and Engineering, Guilin University of Technology, Guilin 541004, China; qtl15697734365@163.com (T.Q.); eydigd@163.com (J.W.)

**Keywords:** chemosensor array, metal ions, fluorescence, carbon quantum dots, amino acids

## Abstract

The simultaneous identification of multiple metal ions in water has attracted enormous research interest in the past few decades. We herein describe a novel method for multiple metal ion detection using a carbon quantum dots (CQDs)-based chemosensor array and the CQDs are functionalized with different amino acids (glutamine, histidine, arginine, lysine and proline), which act as sensing elements in the sensor array. Eleven metal ions are successfully identified by the designed chemosensor array, with 100% classification accuracy. Importantly, the proposed method allowed the quantitative prediction of the concentration of individual metal ions in the mixture with the aid of a support vector machine (SVM). The sensor array also enables the qualitative detection of unknown metal ions under the interference of tap water and local river water. Thus, the strategy provides a novel high-throughput approach for the identification of various analytes in complex systems.

## 1. Introduction

During the past few decades, enormous efforts have been devoted to metal ion detection due to their potential impact on human health and environment. Some metal ions, such as Cu^2+^, Fe^3+^, Zn^2+^, etc., can effectively supplement mineral nutrition and regulate metabolic levels in human body. However, excessive metals can lead to toxicity effects [1,2,3]. Toxic metal ions in drinking water are also a potential problem due to their toxicity and carcinogenicity [4,5]. Therefore, the recognition and differentiation of metal ions is of great significance.

In recent years, many methods for identifying metal ions have been developed. Atomic absorption spectrometry [6], inductively coupled plasma atomic emission spectrometry, and mass spectrometry have been applied [7,8], but these methods require expensive equipment, complex sample preparation process and professional expertise, which limits the low-cost and rapid identification of metal ions. Fluorescence and colorimetry provide a solution to make up for these shortcomings with the advantages of high sensitivity, rapid and high-throughput detection [9,10,11]. However, when faced with the identification of multiple metal ions in a complex system, the interference of co-existing metal ions will cause trouble in the identification of a certain metal ion. Inspired by the mammalian gustatory and olfactory systems [12,13,14], array-based sensor techniques have emerged and attracted widespread interest among scientists as a method to simultaneously detect and identify analytes with similar structures [15,16,17,18]. The sensor array, known as “electronic tongue/nose”, is a pattern recognition-based sensing technology composed of a series of semi-selective sensors with cross-reactive characters [19,20]. The sensor array distinguishes analytes or analyte mixtures through stoichiometric analysis of unique fingerprints. Chemical sensor arrays are becoming increasingly popular due to their advantages in identifying various analytes with high-precision classification [21,22,23,24,25,26]. Recently, chemical sensor array systems have been implemented not only for simultaneous qualitative analysis, but also for quantitative analysis based on pattern recognition algorithms [27,28,29]. Sensor arrays have been reported to identify a large number of analytes, including metal ions [30,31,32], organic compounds and biomacromolecules [33,34,35,36,37,38,39]. Various materials, including metal oxides [40] and semiconductor nanocrystals [41], fluorescent dyes [42], conductive polymers [43] and other nanomaterials [30,44], have been reported to successfully construct sensor arrays. Although these sensing elements are highly sensitive, they are costly or may involve toxicity, which have limitations in obtaining large arrays of sensors for multi-analyte detection. Therefore, simple sensing elements that are capable of constructing chemical sensor arrays are still widely sought after.

Carbon quantum dots (CQDs) have attracted extensive attention in the fields of chemical sensing and biosensing, due to their advantages of ultra-small size, good water solubility, low cell toxicity, tunable surface groups and low cost and easy preparation [45,46,47]. The preparation of CQDs includes top-down and bottom-up methods and can be prepared from various low-cost carbon sources, making CQDs a candidate sensor for the detection of various metal ions [48,49,50,51,52,53,54,55,56,57]. To the best of our knowledge, the design of cross-reaction sensing elements is the core of an array-based sensor system. To further extend the benefits of CQDs, we propose a strategy to easily expand the number of sensing elements by simply self-assembling CQDs with different amino acids without complex chemical synthesis or modification. Jing et al. used an amino acid-modulated inorganic semiconductor quantum dots sensor array to identify metal ions, but toxic semiconductor quantum dots are not the best materials [58]. Wu et al. proposed the synthesis of seven carbon quantum dots as sensing elements to construct sensor arrays, a cumbersome process [44]. In addition, Li et al. obtained two water-soluble fluorescent perylene probes by organic synthesis to form a sensor array for the differentiation of metal ions with a detection limit of 1 μM, but this involved toxic and complex organic synthesis [1]. In this study, five amino acids (L-lutamine, L-histidine, DL-arginine, DL-lysine and L-proline) are self-assembled with CQDs, resulting in six sensing elements (including CQDs). The rich carboxyl and amino groups endow the diversity and intersectionality of the fluorescent sensor array with metal ions. Qualitative and semi-quantitative assays are achieved by linear discriminant analysis (LDA) based on the generation of different fluorescence responses between different sensing elements and metal ions. In addition, regression assay for the quantitative estimation of the concentration of each analyte in the mixture system is implemented using the support vector machine (SVM) algorithm. Unknown metal ions in tap water and river water are identified by Mahalanobis distance analysis (MDA).

## 2. Materials and Methods

### 2.1. Reagents and Equipment

Activated carbon (200 mesh), L-glutamine (Gln), L-histidine (His), DL-arginine (Arg), DL-lysine (Lys), and L-proline (Pro) are purchased from Shanghai Titan Scientific Co., Ltd. (Shanghai, China). MoCl_5_, FeCl_3_·6H_2_O, Cr(NO_3_)_3_·9H_2_O, Ni(NO_3_)_2_·6H_2_O, Cu(NO_3_)_2_·2.5H_2_O, Co(NO_3_)_2_·6H_2_O, Pb(NO_3_)_2_, MnCl_2_·4H_2_O are purchased from Beijing Chemical Reagent Factory (Beijing, China). Er(NO_3_)_3_·5H_2_O, Yb(NO_3_)_3_·5H_2_O, La(NO_3_)_3_·6H_2_O are purchased from Shanghai Macklin Biochemical Co., Ltd. (Shanghai, China). Dialysis membrane (MWCO of 500Da), nitric acid, and sodium hydroxide are purchased from local suppliers. All chemicals are analytical grade and can be used without further purification. River water is collected from local rivers (Li River, Guilin, Guangxi), and tap water is obtained in our laboratory. The ultrapure water used in all experiments is obtained by purification with Asura-AXLM1820 (resistivity of 18.2 MΩ).

The TEM and high-resolution TEM images of AC-CQDs are recorded on 160 kV JEM-2100F (Field emission transmission electron microscopy, JEOL Japan Electronics Co., Ltd). Fourier transform infrared (FTIR) spectra are collected in the range of 500~4000 cm^−1^ on Nicolet Nexux 470 FTIR spectrometer (American Thermoelectric Ltd., West Chester, PA, USA) using the KBr thin section method. X-ray diffraction (XRD) spectra are recorded with PANalytical Empyrean X-Pert PRO (PANalytical B.V.). X-ray photoelectron spectroscopy (XPS) is measured on Escalab 250Xi (American Thermoelectric Ltd) with Al Kα radiation source (hν = 1486.6 eV). Zeta potential measurements are performed on Zeta-sizer 3000HS (Malvern Instruments Ltd. Malvern, UK) by non-invasive backscatter (NIBS) technique. The fluorescence spectra are recorded on Fluoromax-4P spectrophotometer (Horiba, Irvine, CA, USA) with excitation and emission slits set at 1.5 nm, and the luminescence signal is collected at 460 to 750 nm in steps of 1 nm. The decay curves of the sensing elements are recorded on Horiba Fluoromax-4P spectrophotometer equipped with a millisecond lamp. The measurement of the UV-Vis spectra is obtained on Lambda 365 UV-Vis spectrometer (PerkinElmer Instruments Ltd., Waltham, MA, USA). Cell imaging is performed on Laser scanning confocal microscope (Olympus Corporation, Tokyo, Japan). The fluorescence sensor array experiment is conducted on 96-well plates by Biotek SYNERGY H4 microplate reader (BioTek Instruments, Inc., Winooski, VT, USA). 

### 2.2. Preparation of AC-CQDs

AC-CQDs are synthesized using a top-down approach with slight modifications according to previous reports [59]. Briefly, 1 g of dried activated carbon powder (200 mesh) is dispersed into 100 mL of 4 M HNO_3_ solution and refluxed at 120 °C for 24 h. The neutralization of the suspension with NaOH after cooling to room temperature is carried out, followed by centrifugation at 10,000 rpm for 10 min to remove the non-fluorescent material. The supernatant is dialyzed in dialysis bags (MWCO of 500 Da) for 2 days to remove the inorganic salt NaNO_3_. After concentrated freeze-drying, AC-CQDs powders are obtained and dispersed in ultrapure water (0.15 mg/mL) for further use. The fluorescence quantum yield of the AC-CQDs obtained can be found in the Appendix A.

### 2.3. Construction of Sensor Array

To determine the appropriate concentration of amino acids in the construction of carbon quantum dots self-assembled sensor arrays, we titrate AC-CQDs using five different amino acids (NAA) in the concentration range of 0~1132.08 μM and record the changes in fluorescence intensity values ((I_0_ − I)/I_0_) of AC-CQDs at different amino acid concentrations. After determining the final concentration of amino acids in the self-assembled sensor array, the 6 sensing elements (CQDs = sensor 1, CQDs−ln = sensor 2, CQDs−His = sensor 3, CQDs−Arg = sensor 4, CQDs−Lys = sensor 5 and CQDs−Pro = sensor 6) are obtained and successfully constructed a sensor array system.

### 2.4. Experimental Methods for Metal Ion Identification

The sensor array experiments for qualitative and quantitative analysis are performed in 96-well plates, operated by a Biotek SYNERGY H4 microplate reader. Eleven metal ions (Mo^5+^, Fe^3+^, Cr^3+^, Er^3+^, Yb^3+^, La^3+^, Ni^2+^, Cu^2+^, Co^2+^, Pb^2+^ and Mn^2+^) are tested with six sensing elements, respectively, with the following process details: 255 μL of AC-CQD solution (0.15 mg/mL, pH 5.0) is mixed with 30 μL of different amino acids (final concentration of 500 μM), and then 15 μL of metal ion solution is added to the above mixed solution at a certain concentration. All solutions are dispensed at room temperature without contact, and the mixed solution is shaken and allowed to stand for 30 min. Fluorescence emission data from 460 nm to 750 nm are measured and recorded by a Biotek SYNERGY H4 microplate reader. The fluorescence measurements are repeated 24 times for each metal ion concentration, and the obtained spectra data are subjected to Student’s *t*-test to reject 4 abnormal data points, resulting in a coefficient of variation of less than 9% for the final 20 repetitions of the data points and a multi-dimensional response model (6 sensor × 11 metal ions × 20 repetitions). The collected fluorescence change ΔI = I_0_ − I divided by I_0_ (I_0_ and I represent the fluorescence intensity of CQDs/NAA and CQD/NAA/metal-ion solutions, respectively) is normalized and processed to plot the fingerprints of different metal ions. Qualitative and semi-quantitative analysis data are obtained by linear discriminant analysis (LDA) without any processing. Quantitative analysis data is obtained by a support vector machine (SVM) with principal component analysis and automatic scaling preprocessing. The statistical analysis of all data matrices is performed using SYSTAT 13.0 and Solo software.

### 2.5. Analysis of Real Samples

To investigate the capability of the proposed sensor array system in real sample analysis, a blind test is conducted to evaluate the ability of the sensor array to identify unknown metal ions, and tap water and river water are selected as the environmental background for metal ion identification. Tap water is obtained from the laboratory and river water is collected from the Li River (Guilin, Guangxi). Both water samples are centrifuged at 10,000 rpm for 10 min to remove impurities, and the supernatant is filtered through a 0.22 μm membrane and then diluted 10-fold with ultrapure water. The initial eleven metal ions are then randomly assigned to two water samples (with a final concentration of 100 μM) and detected in each of these two environments using the sensor array. The data points of the unknown metal ions are plotted in the LDA diagram, and the type of the unknown metal ion could be defined by comparing the Mahalanobis distance between the unknown metal ion (water sample) and the given metal ion (ultrapure water) in the scatter plot.

## 3. Results and Discussion

### 3.1. Optimization of Detection Conditions

The characterization results of the prepared AC-CQDs are displayed in the Appendix A. In this work, we describe a fluorescence sensor array that can simultaneously identify 11 metal ions and has an excellent classification rate. Fluorescence analysis is based on the aggregation of AC-CQDs induced by the presence of metal ions. The abundance of carboxyl groups, amino groups and hydroxyl groups on the surface of CQDs are prone to self-assembly with different amino acids. In addition, the carboxyl groups of CQDs show fine binding capabilities toward metal ions. As shown in Figure 1a, we constructed a fluorescent sensor array consisting of six sensing elements (one CQD + CQDs × five amino acids). Considering the self-assembly mode as amino acids bound to the surface of AC-CQDs, and some amino acids are free state, we assume that metal ions only bind to AC-CQDs. Under the experimental conditions, we suggest that metal ions and amino acids participate in the aggregation of AC-CQDs (Figure 1b). Amino acids as assemblers confer diversity and detection variability of metal ions relative to AC-CQD-based fluorescent sensor arrays and the discrimination of metal ions shall be possible through the amino acids self-assembled with AC-CQDs (Figure 1c).

We firstly realized the change in fluorescence of CQDs with the self-assembly of five different amino acids, which was confirmed by the experiment results (Figure 2b). The change in the extent of fluorescence is diverse; thus, we can construct a six member sensor array based on AC-CQDs. The fluorescence of AC-CQDs is quenched upon the addition of different metal ions (Figure 3), which, thus, provide a fingerprint-like response pattern for the subsequent multi-dimensional analysis.

In the sensing system, six sensing elements and metal ions act as signal reporters and fluorescence quenchers, respectively. The final concentration of amino acids self-assembled with AC-CQDs is an important parameter affecting the sensitivity of the sensor array. To determine this parameter, we investigate the effect of amino acid concentration on the aggregation of AC-CQDs in the concentration range of 0~1132.08 μM. As shown in Figure 2a, different concentrations of amino acids have little effect on the fluorescence intensity of AC-CQDs, indicating that none of the amino acids cause significant aggregation of AC-CQDs. Therefore, the concentration of amino acids is selected as 500 μM to obtain six sensing elements (Figure 2b). The difference in fluorescence intensity of AC-CQDs caused by the addition of amino acids prove that they both achieved self-assembly. The excitation and emission wavelengths of the six sensing elements are the same, 450 nm and 524 nm, respectively, and the quantum yields are shown in Appendix A. In addition, we investigate the stability of the six sensing elements within 72 h. As shown in Appendix A, it can be observed that the fluorescence intensity of CQDs-Arg and CQDs-Lys decrease in the first 6 h, while the remaining four sensing elements maintain good fluorescence stability, which may be related to the chain length of amino acids. Before performing the array detection, we investigate the incubation time of the metal ions with the sensing elements. Pb^2+^ is chosen as representative metal ion, and Appendix A shows the fluorescence kinetic curves of six sensing elements after the addition of Pb^2+^ (100 μM). It can be observed that the fluorescence intensity basically reaches stability after 30 min, indicating that the reaction between the metal ion and the sensing element reaches dynamic equilibrium. Most metal cations form hydroxide precipitates under neutral and alkaline experimental conditions; therefore, neutral and alkaline experimental conditions are not suitable for our work. Considering this factor, pH 5.0 is selected as the optimal pH in all array experiments [60].

### 3.2. Metal Ion-Induced Sensing Element Aggregation

We investigated the effect of eleven metal ions on the induced aggregation of six sensing elements. The fluorescence titration experiment was carried out at room temperature to study the quenching behavior of metal ions (0~614.04 μM) on the sensing elements, and the titration curve of fluorescence intensity was obtained. As shown in Figure 3, most metal ions reach about 200 μM and the curve starts to flatten, indicating that the reaction between the metal ions and the sensing element tends to be in dynamic equilibrium. The Stern–Volmer equation for 11 metal ions is established as follows:$$I_{0}/I=1+K_{\mathrm{SV}}\left[Q\right]$$
where $I_{0}$ and I are the fluorescence intensity in the absence or presence of metal ions, and $K_{\mathrm{SV}}$ is the quenching constant for the quenching efficiency of metal ions and $\left[Q\right]$ is the concentration of metal ions [48]. The calculated Stern–Volmer constants are shown in Appendix A. We can observe through the changes in fluorescence intensity that the presence of different metal ions at the same concentration can induce differences in the aggregation of the same sensing element. For example, for CQDs-Gln, the fluorescence intensity quench of Fe^3+^ on the sensing element is much greater than that of Mn^2+^, and for CQDs-His, the quench of Fe^3+^ and Mn^2+^ on the sensing element is comparable. The degree of fluorescence intensity quench varies for all six sensing elements as the metal ion concentration increases. On the other hand, the induced aggregation of the same metal ion on different sensing elements shows significant differences, for example, in Figure 3, the fluorescence quench of Mn^2+^ at 200 μM is significant for CQDs-Arg and insignificant for CQDs, which is attributed to the high affinity of Mn^2+^ and arginine [61].

Based on the above conclusions, we learn that carboxyl, amino and hydroxyl can react with metal ions to form complexes, and the difference in binding strength is related to the specific sensing element and metal ions used. We chose Pb^2+^ as a representative to study the induced aggregation of metal cations on sensing elements and the results show that Pb^2+^ can effectively induce the aggregation of sensing elements. The fluorescence spectra of the six sensing elements at different concentrations of Pb^2+^ are shown in Appendix A and the presence of Pb^2+^ with different concentrations can cause changes in the fluorescence intensity of the sensing element. The absence of Pb^2+^ can be observed on the TEM images and the sensing elements are well dispersed on the transmission electron microscope grid with sizes of 2.85, 3.47, 4.33, 4.41, 4.35, 6.37 nm, respectively. However, in the presence of Pb^2+^, the sensing elements complexed with Pb^2+^ and formed large clusters with sizes all larger than 50 nm, indicating that Pb^2+^ induced sensing element aggregation (Appendix A). The UV–Vis absorption spectra show significant changes, with the absorption intensity enhanced by electron transitions due to the aggregation of sensing elements (Appendix A). The fluorescence lifetime and Zeta potential (Appendix A) show similar quenching patterns in the presence and absence of Pb^2+^, corresponding to similar lifetime and Zeta potential values, indicating that static quenching occurred in the sensor when metal ions induced sensing element aggregation. Moreover, the FTIR spectra prove that Pb^2+^ is successfully combined with the sensor element. As observed in Appendix A, the presence of Pb^2+^ leads to a change in the intensity of the IR peak, which is attributed to the change in the stretching vibration of OH [62]. These results indicate that the induced aggregation of metal ions can be understood as the complexation of metal ions with AC-CQDs and the coordination of amino acids. The interactions between sensing elements, metal ions and amino acids integrate electrostatic and coordination effects to induce static quenching of surface traps or electron–hole complexes through electron or energy transfer processes.

### 3.3. Response of the Sensor Array to Metal Ions

Initially, we select the appropriate concentration of metal ions near the inflection point of the fluorescence titration curve to obtain the sensitive detection results. The maximum fluorescence emission wavelength (524 nm) of each sensing element does not shif at 450 nm excitation (Appendix A). By integrating the change in fluorescence intensity ((I_0_ − I)/I_0_) before and after adding the metal ion solution, a fluorescence pattern belonging to each metal ion is obtained (Figure 4), also known as a metal ion fingerprint (6 sensing elements × 11 metal ions × 20 repetitions). The result shows the difference in binding between the different sensing elements and metal ions under the modulation of amino acids, confirming the feasibility of using this sensor array to identify multiple metal ions.

In sensor arrays, cross-reactivity between the sensing elements and analytes is critical for the classification rate of analytes in the array. Chemical sensor arrays are constructed through the nonspecific reactions accumulated by crosstalk between different sensing elements and analytes to simultaneously identify multiple analytes qualitatively and quantitatively. The multi-responsiveness of self-assembled sensor systems encourages the fabrication of metal ion sensor arrays. In the qualitative analysis, linear discrimination analysis (LDA) is used for the training matrix (6 sensing elements × 11 metal ions × 20 repetitions). LDA is a supervised method in pattern recognition algorithms that enables the dimensionality reduction and classification of multivariate data. In addition, we evaluate the level of correct classification of clusters in the observations by the leave-one-out method (a cross-validation method, also known as the Jackknife method). As shown in Figure 5a, the three-dimensional LDA plot obtained from the first three factors (F1 = 97.9%, F2 = 1.8%, and F3 = 0.2%) shows that the 11 metal ions and the control (a total of 240 data points) are clearly distinguished from each other and that twenty replicate trials of the same metal ion are essentially concentrated at one point, demonstrating that LDA maximizes the separation between multiple analytes, while minimizing the separation between the replicate measurements of the same analyte [58]. Meanwhile, the classification rate of the results obtained from the output of the leave-one-out method for the 11 metal ions and control is 100% (Appendix A). We try to simplify the sensor array by using fewer sensing elements, selecting sensor 1, sensor 2 and sensor 6 as the components of the sensor array. From the LDA plots drawn for the first three factors, it can be observed that the eleven metal ions and the control are also distinguished from each other, but there is a small difference between twenty repetitions of the same metal ion (Figure 5b), indicating that the high sensitivity of the sensor array depends heavily on the number of sensing elements. Based on the above results, the fluorescence sensor array constructed by self-assembly of amino acids and AC-CQDs can be an effective means to simultaneously recognize and distinguish different metal ions in water without complex chemical modification.

After successfully identifying 11 metal ions, we perform the next challenging experiment to determine whether the sensor array can recognize different concentrations of metal ions, which may lead to drastic changes in the fluorescence response pattern, as well as affect the application in the real samples. Metal ions are present in the ecological environment as a mixture, and their mixed toxicity has a serious impact [63]; therefore, we decide to use the designed sensor array for quantitative analysis. Three typical metal ions (Pb^2+^, Mn^2+^ and Cr^3+^) in electroplating wastewater are chosen as representatives. First, to evaluate the ability of the sensor array as a monitoring tool for metal ion levels in water, we use LDA for semi-quantitative analysis. As shown in Figure 6a, it can be observed that the individual concentrations of different metal ions are well separated with 100% classification rate (Appendix A). In addition, the three-dimensional LDA plot shows that the different concentrations corresponding to these three metal ions are distributed in a line for each, which is consistent with the fluorescence titration results (Appendix A). Meanwhile, simplified sensor arrays (sensor 1, sensor 2 and sensor 6) are also used for semi-quantitative analysis. As can be observed in Figure 6b, the sensitivity of the simplified sensor array is lower than that of the original sensor array, and greatly different from the result of fluorescence titration. The linear relationship of the concentration response curve shows that the interaction between different concentrations of metal ions and the sensor element is uniform and stable, indicating that the sensor array has high repeatability.

Next, we try to perform regression analysis on the mixture of metal ions. The support vector machine (SVM) algorithm, [64] suitable for complex nonlinear response, is used to quantitatively estimate the concentration of various metal ions in the mixed system. Figure 7 shows the results of the quantitative array for the mixture of Pb^2+^, Mn^2+^ and Cr^3+^ at the same concentrations. Here, we execute the SVM algorithm through the Solo software to classify the data obtained by 11 kinds of mixed concentrations, partly for calibration and model development, and partly used as the unknown concentration for cross-validation (first 40% and last 20% of the whole data, red circles). Good root-mean-square error values (RMSE) indicate that the SVM shows accurate classification and good predictive ability for the constructed model, demonstrating that the self-assembled sensor array based on amino acids and AC-CQDs can effectively predict the concentrations of Pb^2+^, Mn^2+^ and Cr^3+^ in the mixed system, which is expected to provide researchers with a novel approach to environmental hazard assessment.

### 3.4. Determination of Unknown Metal Ions in Real Samples

To explore the utility of self-assembled sensor arrays of amino acids and AC-CQDs in identifying unknown metal ions in different water samples, a blind experiment is conducted here. The unknown metal ions in tap water and river water are identified by the the sensor array system (6 sensing elements × 11 metal ions × 20 repetitions) using tap water and river water as background environments, respectively. The two water samples are pretreated according to the above method, and then unknown metal ions with a final total concentration of 100 μM are added to the different water samples for the experiments, and the fluorescence response of the sensor array to the unknown metal ions in tap water and river water is collected. The data obtained are analyzed by LDA, the first two factors are plotted as shown in Figure 8, and the different fluorescence response patterns generated by the sensor array are obtained (Appendix A). Unknown metal ions in both tap water and river water samples can be identified according to the Mahalanobis distance analysis (MDA), which proves that the testing accuracy of the sensor array is not affected by the complex testing environment with the recognition rate of 100%.

## 4. Conclusions

In summary, we successfully prepare AC-CQDs from activated carbon, and design and construct a fluorescent sensor array based on the self-assembly of carbon quantum dots. Amino acids are combined with AC-CQDs as the assembling agent, resulting in the sensing elements’ differentiation. Based on the static quenching of aggregation due to the electron transfer between metal ions and sensing elements, 11 metal ions are successfully identified by LDA with 100% classification rate. In addition, the quantification of some metal ions is successfully achieved. The sensor array also enables the qualitative identification of unknown metal ions in specific environments. Combined with the ease of design and construction of this self-assembled sensor array, we believe that the method is versatile and can easily expand the number of sensing elements without complex and time-consuming chemical synthesis processes, and allow researchers to experiment with changing the subject (sensing element) and object (analyte) according to the purpose.

## Figures and Tables

**Figure 1 molecules-27-03843-f001:**
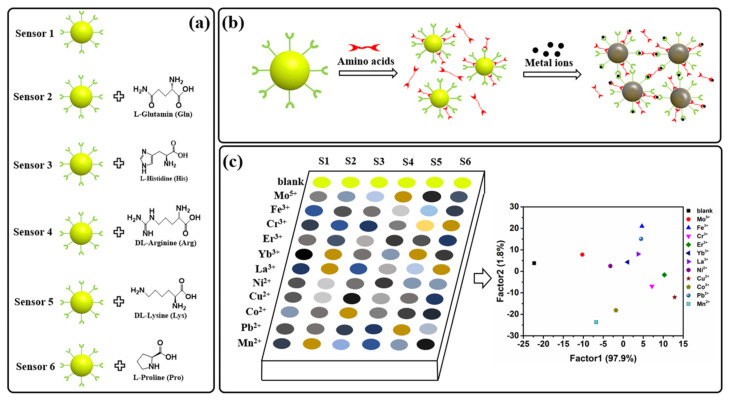
(**a**) Schematic structure of the nanoparticle sensors (S1–S6) constructed with the self-assembled of amino acids and carbon quantum dots (AC−CQDs); (**b**) The feasible colorimetric detection mechanism for the targets; (**c**) Diagram of colorimetric sensor array and detection principles of metal ions.

**Figure 2 molecules-27-03843-f002:**
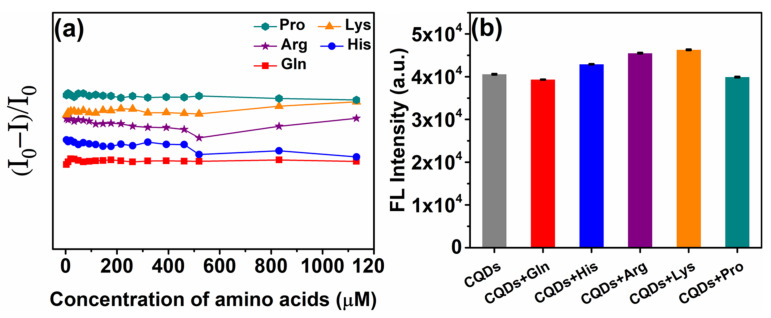
(**a**) In the absence of metal ions, the fluorescence response (I_0_ − I)/I_0_ value of AC-CQDs depends on the amino acid concentration (I_0_ is the emission intensity of AC-CQDs at 524 nm in the absence of amino acids. I is the emission intensity of AC-CQDs mixed with various amino acid solutions at 524 nm). (**b**) In the presence or absence of amino acids, the emission intensity of AC-CQDs at 524 nm. (AC-CQDs) = 0.15 mg/mL, (amino acids) = 500 μM.

**Figure 3 molecules-27-03843-f003:**
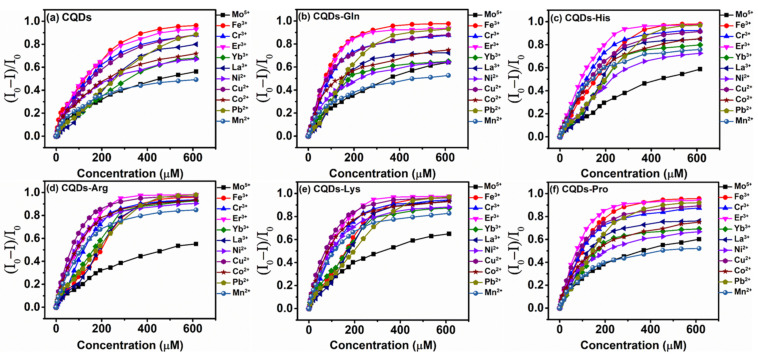
Effect of 11 metal ions at different concentrations (0~614.04 μM) on the fluorescence response of six self-assembled sensing elements (**a**–**f**). I and I_0_ are the emission intensity collected at 524 nm (λ_E*x*_ = 450 nm) in the presence or absence of metal ions (Room temperature, pH 5.0).

**Figure 4 molecules-27-03843-f004:**
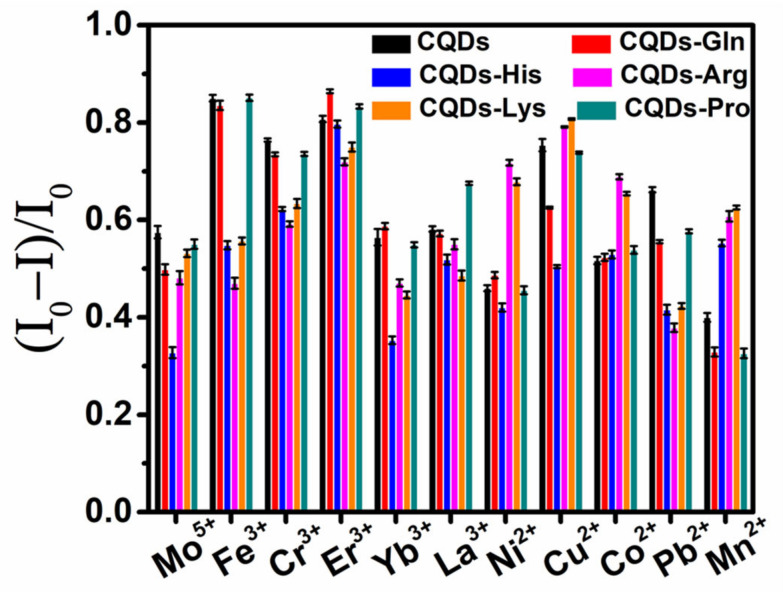
Fingerprints generated by the carbon quantum dot-based self-assembled sensor array for 11 metal ions (200 μM). Error bars represent the standard deviation of each sensing element–metal ion pair for 20 replicate tests.

**Figure 5 molecules-27-03843-f005:**
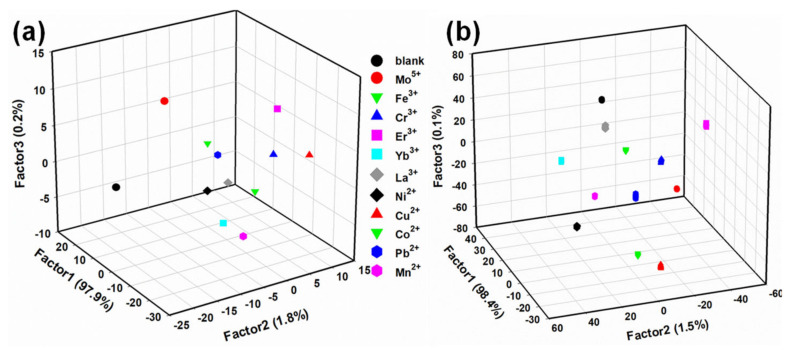
The three−dimensional standard scores of the fluorescent sensor array for eleven metal ions and the blank control obtained from LDA. (**a**) Original fluorescence sensor array. (**b**) Simplified fluorescent sensor arrays (sensor 1, sensor 2 and sensor 6). (Metal ion) = 200 μM, (Amino acid) = 500 mM, with 20 repetitions for each metal ion. The cross-validation procedure shows 100% correct classification.

**Figure 6 molecules-27-03843-f006:**
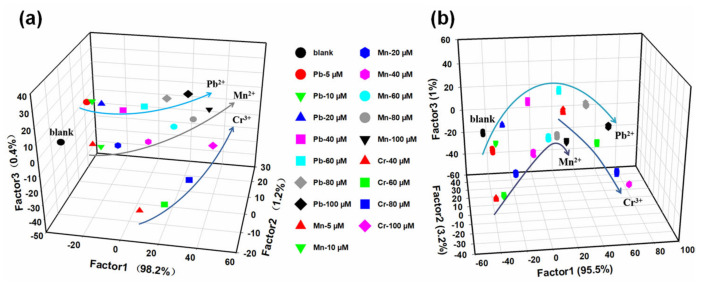
Semi−quantitative analysis results of three metal ions at different concentrations obtained from LDA. (**a**) Original fluorescence sensor array. (**b**) Simplified fluorescent sensor arrays (sensor 1, sensor 2 and sensor 6). For both sensor array systems, (Pb^2+^ and Mn^2+^) = 5~100 μM and (Cr^3+^) = 40~100 μM, with 20 repetitions for each concentration.

**Figure 7 molecules-27-03843-f007:**
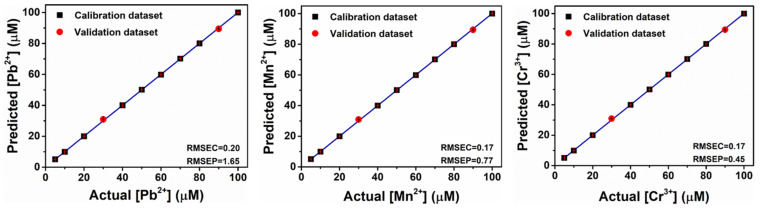
Regression analysis results of the concentrations of metal ion mixtures Pb^2+^, Mn^2+^ and Cr^3+^ in the quantitative analysis. The data are obtained by the fluorescence sensor array. Detailed mixing concentration conditions are displayed in the Appendix A. For multiple concentrations of each metal ion, the root-mean-square error (RMSE) values and plots of calibration (C) and prediction (P) demonstrate the high accuracy of the model and prediction.

**Figure 8 molecules-27-03843-f008:**
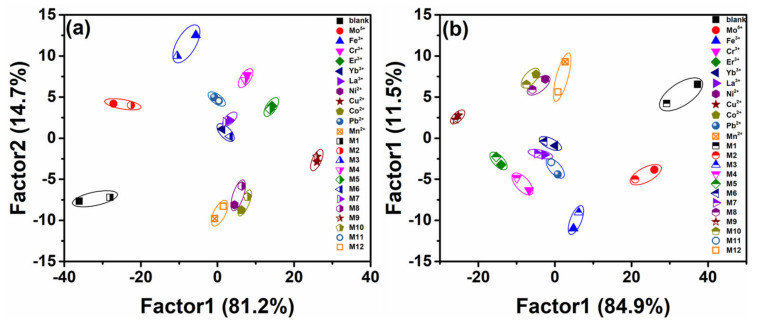
In the simulated real environment, unknown metal ions are identified by Mahalanobis distance using fluorescence sensor array. M is an unknown metal ion randomly selected from the original 11 metal ions. (**a**) Laboratory tap water, (**b**) river water.

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
