# Peer review of "Carbon Quantum Dots Based Chemosensor Array for Monitoring Multiple Metal Ions"

_molecules, 2022, doi:10.3390/molecules27123843_

Round 1

Reviewer 1 Report

This work presented the method for multiple metal ions detection using a carbon quantum dots (CQDs) based chemosensor array by functionalization with different amino acids. Although the method of preparation of these sensors is not new, the implementation to chemosensor array is well presented. The CQDs with and without metal ions were systematically characterized (TEM, IR, XRD, XPS, etc). It is also impressive that all 11 metal ions were successfully identified by LDA with 100% classification rate. The manuscript is well written. So, I suggest to accept this paper to Molecule with only minor revision:

1.       In page 2, you write the heading “2. Results”, but the content of that paragraph belong to the introduction part. So, please remove the heading “2. Results”.

Author Response

We would sincerely thank the Review for the time and effort in carefully reading the manuscript and in preparing the review reports. We truly appreciate your positive comments on our work, as well as for raising interesting points, which lead to the improvement of the manuscript. We have revised the manuscript accordingly. The point-by-point responses to the comments are enclosed. We hope we have satisfactorily addressed all the concerns and questions.

Reviewer 2 Report

The manuscript  is an interesting research about developing a chemosensor array for metals ions detection simultaneously. It is very well written and presented. A detailed supllementary material is very helfup and useful for readers.
Here are some comments:
line. 57-58. It seems that Figure 1 should be placed in other section, namely Section 3.1. , where it is mentioned for the first time (line 176).

Line 59. Remove "2. Results"

line 148. What does NAA mean?

line 251. Write "On TEM images " instead of  "Transmission electrone microscopy can be observed..."

line 284. Remove the word "that"
